# Cardiovascular, Hemodynamic, and Anthropometric Adaptations Induced by Walking Training at FATmax in Obese Males and Females over 45 Years Old

**DOI:** 10.3390/ijerph22050701

**Published:** 2025-04-29

**Authors:** Laurence Mille-Hamard, Iman Momken, Jean-Pierre Koralsztein, Véronique Louise Billat

**Affiliations:** 1Unité de Biologie Intégrative des Adapations à l’Exercice (UBIAE), EA 1374, Université d’Evry-Paris Saclay, 91000 Evry-Courcouronnes, France; iman.momken@univ-evry.fr (I.M.); veroniquelouisebillat@gmail.com (V.L.B.); 2Informatique, Bio-Informatique et Systèmes Complexes (IBISC), EA 4526, Université Paris-Saclay, 91020 Evry, France; 3INSERM UMR-S 1180, Faculté de Pharmacie, Université Paris-Saclay, 91400 Orsay, France; 4Billatraining, 91840 Soisy sur Ecole, France; jean.pierre.koralsztein@orange.fr; 5EA 4445-Movement, Balance, Performance, and Health Laboratory, Université de Pau et des Pays de l’Adour, 65000 Tarbes, France

**Keywords:** individualized exercise, obesity, cardiovascular health

## Abstract

Aims and Objectives: The present study aimed to examine the effects of 6 months of unsupervised training, walking at maximal fat oxidation (FATmax), on body composition and cardiovascular function at rest and exercise, in middle aged obese subjects. Methods and results: A single group with pre-test/post-test study design was conducted. Eighteen obese subjects (11 males and 7 females) over 45 were engaged in a non-supervised walking training for 6 months, 40 min, 3 times per week, at the targeted HR corresponding to FATmax (5.5 ± 0.6 km·h^−1^). This training modality led to a reduction in obesity-related indicators among participants, including weight (−3.7 ± 3.4 kg), BMI (−1.4 ± 1.3 kg/m^2^), waist circumference (−5.6 ± 4.7 cm), and body fat percentage (−2.1 ± 2.7%). However, we observed a great variability in this response to training according to individuals. Furthermore, heart rate and rate of pressure product (RPP) at rest significantly decreased (6% and 11% respectively) as well as the cardiac load during exercise (RPP −11% and cardiac cost −8%) after training. In conclusion, walking at FATmax is an efficient non-supervised training modality, allowing improvement in both body parameters and cardiovascular markers at rest and during exercise in middle age obese subjects. Even if body parameter changes were modest, the cardiac load decrease is an important factor for reducing the risk of cardiovascular diseases in this population.

## 1. Introduction

The prevalence of obesity has risen dramatically over the past few decades, reaching epidemic proportions worldwide. By 2022, more than one billion people were classified as obese, more than doubling in prevalence among adults since 1990 [1]. Obesity is now widely recognized as a chronic condition that contributes to numerous health risks. In France, the rate of obesity among older adults has increased significantly, with 19.9% of individuals over the age of 65 being classified as obese in 2020, compared to a national average of 17% among all adults [2]. The World Health Organization defines obesity as an excess accumulation of body fat—greater than 25% in men and 30% in women—that adversely affects health and is closely linked to premature cardiovascular disease (CVD). The relationship between obesity and cardiovascular health is complex, as obesity encompasses both “metabolically healthy” and “unhealthy” phenotypes. The metabolically healthy obesity phenotype is characterized by the absence of metabolic abnormalities despite excess body fat. However, this status is dynamic, with a tendency to transition to a metabolically unhealthy state, especially with aging, leading to increased risks of CVD. Furthermore, research indicates that individuals with metabolically healthy obesity may face a higher risk of developing CVD at a younger age compared to those with normal weight [3,4].

Obesity and hypertension (HTN) are closely intertwined conditions, with excessive adiposity being a major contributor to elevated blood pressure (BP) and the development of HTN, particularly in middle-aged and elderly populations [5]. HTN and obesity are associated with higher risk of CVD. Lowering systolic blood pressure (SBP) by 10 mmHg can reduce the risk of CVD by 29% [6], illustrating the significant impact of BP management in reducing cardiovascular risk. Furthermore, elevated diastolic blood pressure (DBP)—particularly isolated DBP, which reflects increased peripheral resistance—has emerged as an underrecognized risk factor for cardiovascular disease. Hypertension-mediated organ damage has been observed in patients with isolated diastolic hypertension [7,8]. In obese individuals, changes in body weight are strongly correlated with changes in BP [9,10]. Weight loss, either through dietary changes, physical activity, or surgical interventions, has been shown to reduce myocardial oxygen consumption and improve cardiac function, as evidenced by decreases in SBP, heart rate (HR), and rate of pressure product (RPP), an indicator of cardiac workload [11,12,13]. Under resting conditions, safer RPP should range between 7000 and 9000 mmHg/min. A alue above 10,000 mmHg/min indicates an increased risk for heart disease. Furthermore, an excessive RPP during exercise may causes myocardial ischemia due to a mismatch between myocardial oxygen demand and supply [14,15].

Exercise is a cornerstone of interventions aimed at managing obesity and HTN. Beyond facilitating weight loss through an improved energy balance, exercise exerts profound effects on cardiovascular function through mechanisms like enhanced endothelial function, reduced inflammatory cytokines, and positive epigenetic changes [16]. Endurance training, in particular, has been shown to reduce resting BP, with reductions of approximately 2.95 mmHg in SBP and 1.93 mmHg in DBP observed across various populations [17]. For overweight or obese individuals, training that targets fat oxidation can be particularly effective. This approach, known as training at FATmax or Lipoxmax, focuses on exercise intensities that maximize fat utilization during activity [18]. Training at FATmax, typically a low- to moderate-intensity exercise is accessible for overweight and obese individuals and has been shown to improve body composition by promoting fat loss while preserving lean mass [19,20]. Studies have also reported improvements in lipid profiles and modest reductions in resting BP following FATmax training [21,22]. One study conducted with non-obese women showed improvements in cardiovascular function after supervised FATmax training [23]. However, very few studies have focused on cardiovascular adaptations including cardiac load during exercise. This is an important issue for a population of obese people, whose increased BP at rest as well as during exercise can negatively affect cardiac work. Most studies to date have utilized supervised exercise protocols involving running or cycling [19,21,22,24], with limited research on the cardiovascular effects of non-supervised walking at FATmax, a more feasible and sustainable form of physical activity for older adults.

Walking offers a practical and low-impact option for maintaining physical activity in everyday life, especially for those who may be deterred by the demands or injury risks of higher-intensity exercise. Given the gaps in the existing literature, this study aims to evaluate the effects of a 6-month non-supervised walking program at FATmax intensity on body composition and cardiovascular parameters in obese men and women over the age of 45. Specifically, the study tests the hypotheses that (1) walking at FATmax over 6 months will lead to significant reductions in body weight and fat mass, and (2) the program will result in improved cardiac load, as indicated by reductions in RPP and other cardiovascular markers at rest and during exercise.

## 2. Methods

### 2.1. Study Design and Participants

This study employed a longitudinal pre-test/post-test design without a control group. Participants were recruited from the “Caisse Centrale des Activités Sociales medical center” in Paris (CCAS), France. Inclusion criteria included age over 45 years, a fat mass percentage above 25% for men and 30% for women, and the ability to walk three times weekly for 30–40 min. Exclusion criteria included engagement in regular exercise or weight loss programs within the last six months, cardiac or respiratory disorders, and the use of medication that could interfere with the cardiovascular or metabolic outcomes of interest.

Eighteen subjects (11 males and 7 females) participated, with an average age of 50.9 ± 5.7 years (49.6 ± 4.7 for men and 51.9 ± 2.2 for women). The average body mass index (BMI) was 31.8 ± 3.8 kg/m^2^. Among the seven female participants, five were postmenopausal. Ethical approval was obtained from the institutional review board, and written informed consent was provided by all participants.

### 2.2. Assessment and Measurements

Body Composition: Body weight, height, waist circumference (WC), and skinfold thickness (Harpenden, UK) were measured at baseline (D1), 3 months (D2), and 6 months (D3). Fat mass (FM) percentage was estimated using the skinfold thickness method and the Siri equation [25]. BMI was calculated as weight (kg) divided by height (m^2^).

Cardiovascular Function: Resting SBP and DBP were measured using a calibrated mercury sphygmomanometer after 10 min of seated rest. Resting HR was also recorded.

### 2.3. Graded Exercise Test (GXT)

GXT was conducted every three months to assess oxygen consumption (VO_2_) and carbon dioxide production (VCO_2_), using a Jaeger Oxycon Pro system (Erich Jaeger GmbH, Hoechberg, Germany). Participants performed a GXT on a treadmill with increments of 0.5 km/h every 6 min until volitional fatigue. They were instructed to walk. VO_2_peak was calculated based on breath-by-breath analysis of VO_2_. Cardiac output (Qc), stroke volume (SV), and HR were continuously measured using Physioflow technology (Manatec, Poissy, France). Blood pressure was measured at each stage.

### 2.4. Calculations

FATmax was identified through analysis of steady-state fat oxidation rates during submaximal stages of the GXT. The fat oxidation rates were calculated from the gas exchange measurements, during the last minute of each stage of the GXT, according to the non-protein respiratory quotient technique using the following equation: Fat (mg·min^−1^) = 1.6946 × VO_2_ − 1.7012 × VCO_2_ (gas volume expressed in mL·min^−1^) [26].

The RPP was calculated as HR × SBP, at rest and during the last minute of each stage of the GXT. The cardiac cost (CC) was calculated as HR/velocity (with the unit beat per meter) at rest and during the last minute of each stage of the GXT. A mean value for RPP and CC was calculated for the whole exercise, before and after the training period.

### 2.5. Training Protocol

The walking program was individualized based on each participant’s FATmax, determined through the GXT on a treadmill. FATmax represented the exercise intensity at which the greatest rate of fat oxidation occurs. Participants engaged in a 40 min walking session three times a week, targeting the HR corresponding to their FATmax, which averaged 112 ± 12.6 bpm (approximately 67% of their theoretical HRmax). This HR corresponded to a walking speed of 5.5 ± 0.6 km/h at baseline.

Participants used a Polar FT1 (Polar, Kempele, Finland) heart rate monitor during training to ensure adherence to the target HR range (±5 bpm). Training sessions were unsupervised but included weekly phone check-ins to promote adherence and resolve any issues. Participants were instructed to avoid vigorous physical activity outside of the prescribed sessions and to maintain their usual diet.

### 2.6. Statistical Analysis

Quantitative variables are presented as mean ± standard deviation (SD). Statistical analysis was performed using SPSS version 27.0. Normality was assessed using the Shapiro–Wilk test. Repeated measures ANOVA were used for comparisons of pre- and post-training outcomes normally distributed. Post hoc comparisons following significant ANOVA results were adjusted using Bonferroni correction. Statistical significance was set at *p* < 0.05. Effect sizes were calculated using Cohen’s d to determine the magnitude of change, with d ≥ 0.80 interpreted as a large effect, 0.5 < d < 0.8 interpreted as a medium effect, and 0.2 < d < 0.5 interpreted as a low effect [27]. The threshold value for the interpretation of a large ES was ≥ 0.80.

## 3. Results

### 3.1. Body Composition

The 6-month walking program resulted in significant reductions in body weight (−3.8 ± 3.4 kg, *p* < 0.001), BMI (−1.4 ± 1.3 kg/m^2^, *p* < 0.001), and WC (−5.6 ± 4.7 cm, *p* < 0.001). Fat mass decreased significantly (−3.0 ± 2.9 kg, *p* < 0.01), while lean body mass remained stable, indicating that weight loss was primarily due to fat loss (Table 1). Despite inter-individual variability (Figure 1), five of the eighteen participants (three women, two men) achieved clinically significant weight loss (≥5%), and four men ended the training session with a non-obese status (FM% below 25).

### 3.2. Exercise Performance

Maximal walking speed during GXT increased significantly, from 6.2 ± 0.5 km/h at baseline to 6.5 ± 0.5 km/h at 6 months (*p* = 0.013), indicating improved aerobic capacity. However, VO_2_peak increased modestly from 22.3 ± 3.8 mL/kg/min to 23.6 ± 4.1 mL/kg/min, though this change did not reach statistical significance. The speed corresponding to FATmax has been increased to 5.8 ± 0.6 km·h^−1^ after training (*p* = 0.034), while the targeted HR, i.e., the HR associated with maximal lipid oxidation, did not change across training (*p* > 0.05). Furthermore, the training period ended without injury.

### 3.3. Cardiovascular Adaptations

Resting Cardiovascular Measures: Resting SBP decreased from 131 ± 14 mmHg at baseline to 123 ± 14 mmHg after 6 months (*p* = 0.012), while DBP decreased from 100 ± 10 mmHg to 95 ± 6 mmHg (*p* = 0.014) (Table 2). Resting HR showed a non-significant reduction overall (77 ± 13 bpm to 73 ± 11 bpm, *p* = 0.09), but significantly decreased in female participants (*p* = 0.007). Meanwhile the resting RPP decreased significantly and the decrease attained significant level at D3 compared with D1 (*p* = 0.004); six subjects thus showed resting RPP > 10 000 mmHg·min^−1^ after training compared to eight at the first visit. This reduction in resting RPP suggested a lower myocardial oxygen demand following training.

Exercise Cardiovascular Measures: The average RPP during exercise decreased by 11% from baseline to 6 months (*p* = 0.0004), reflecting reduced cardiac oxygen consumption during physical exertion (Figure 2). The decrease was significant in men (*p* = 0.02) as well as in women (*p* = 0.03) (Table 2). In the meantime, the cardiac cost of walking decreased with training (8%, *p* = 0.0001) (Figure 3). We failed to correlate the RPP or CC decrease with the weight loss. The decrease in RPP was associated with a decrease in the HR (*p* = 0.022) and SBP (*p* = 0.001) during exercise (Figure 2). Conversely, Qc and SV remained stable throughout the study protocol (Table 2), suggesting that cardiovascular improvements were more likely related to peripheral vascular adaptations rather than central cardiac changes.

## 4. Discussion

This study provides evidence that a 6-month non-supervised walking program at FATmax can elicit significant improvements in body composition and cardiovascular health in obese individuals over 45 years old. The observed reductions in cardiac cost, RPP, and body weight, particularly fat mass, suggest that such an exercise regimen can serve as a viable intervention for reducing cardiovascular risk factors in this population.

### 4.1. Benefits of FATmax Training and Body Composition

FATmax typically occurs at a low to moderate exercise intensity, making it accessible and sustainable for overweight or obese individuals, particularly those who are older and may have joint or cardiovascular limitations. This intensity corresponds to around 40–45% of VO_2_max in men and women, which is sufficient to induce weight loss and promote adaptations in fat metabolism without inducing excessive strain [20,28,29]. Furthermore, FATmax training aligns with the concept of “metabolic fitness”, as it enhances the body’s ability to oxidize lipids, which is crucial for reducing visceral adiposity and improving metabolic health [24,30]. Indeed, visceral fat is closely associated with inflammation leading to increased cardiovascular risk [31]. By targeting the intensity at which fat oxidation is maximized, FATmax training helps to mobilize and utilize fatty acids from adipose tissue more efficiently. The results from this study, showing significant reductions in fat mass and WC, underscore the effectiveness of this approach in reducing adiposity, especially visceral fat. The preservation of lean body mass observed in this study is consistent with findings from other studies employing FATmax training [18,20]. This preservation is crucial for older adults, as maintaining muscle mass is key to sustaining functional capacity and metabolic rate [32]. By focusing on FATmax, the current study confirmed a method that balances weight loss with muscle preservation, which is particularly advantageous in the context of aging.

### 4.2. Cardiovascular Adaptations Through FATmax Training

One of the significant findings of this study was the 11% reduction in RPP both at rest and during exercise. RPP, calculated as the product of HR and SBP, serves as a non-invasive indicator of myocardial oxygen consumption [33]. This lower RPP values is thus a critical factor in managing cardiovascular health in obese individuals. The lowering of CC induced by the training session confirmed the lesser work of the heart during exercise. A lower RPP at rest and during exercise was also recorded in young obese men after combined exercise training (aerobic and resistance training). It was suggested that this decrease in RPP was related to a decrease in systemic vascular resistance [34]. In the present study, despite moderate exercise intensity and the middle age of the subjects, the training sessions allowed the heart workload at rest and during exercise to decrease significantly. Furthermore, the lack of correlation between the RPP decrease and the loss of body weight indicated that the reduction in RPP was due to exercise per se, rather than the weight loss. Indeed, reduction in resting RPP was recorded also without weight loss after aerobic training in sedentary men [14]. A SBP decrease around 8 mmHg at rest and 9 mmHg during exercise partly mediated the decrease in RPP. It was noted, however, that a decrease in SBP of 10 mmHg is associated with a 29% decrease in the risk of CVD [6]. Hypertension has a multifactorial etiology and, therefore, different mechanisms are involved in the hypotensive effects of endurance training. The mechanisms behind the reduced RPP, mediated by reduced BP observed with our FATmax training, may involve improvements in endothelial function and reduced arterial stiffness, both of which are positively influenced by regular moderate-intensity exercise [35]. Endurance exercise is known to enhance nitric oxide availability, which helps to relax blood vessels and improve vascular compliance, thereby reducing peripheral resistance and lowering SBP. These changes, combined with the reduced HR measured during exercise, contribute to a more efficient cardiovascular system both at rest and during physical exertion [34,36,37,38]. Even though the training intensity was quite low in the present study, it was sufficient to promote adaptations usually described after endurance training that represent health promoting adaptations. FATmax training’s emphasis on maintaining exercise at a submaximal intensity may also play a role in avoiding the exaggerated hypertensive responses to exercise that are sometimes observed in higher-intensity activities. Previous studies have shown that moderate-intensity aerobic exercise can lower both resting and ambulatory BP in hypertensive and normotensive individuals [17,37]. The present study extends these findings by demonstrating that similar benefits can be achieved through a non-supervised walking program that focuses on FATmax, suggesting that the cardiovascular adaptations seen with more structured programs can be replicated in a more practical, everyday context.

### 4.3. The Role of Autonomy in FATmax Training

A notable strength of the current study is its non-supervised design, which highlights the feasibility of applying FATmax training in real-world settings without the need for constant supervision. This aspect is particularly relevant for public health, as it suggests that individuals can achieve significant health benefits through relatively simple guidance on walking at the appropriate intensity using a heart rate monitor. The non-supervised nature of this program also addresses barriers to exercise adherence, such as time constraints and access to facilities, which are common challenges for middle-aged and older adults [39]. The fact that participants in this study maintained the prescribed exercise intensity using heart rate monitors indicates that FATmax training can be effectively self-managed. This aligns with recent research suggesting that individualized, heart-rate-based prescriptions can improve adherence to exercise programs [40]. The simplicity of walking as a mode of exercise further supports the implementation of FATmax training, as it does not require specialized equipment or facilities, making it accessible even to those with limited resources or mobility.

However, the variability in individual responses observed in this study, as in other studies [23,41], also highlights the importance of regular re-assessment and individualized adjustments. Some participants experienced more significant weight loss and cardiovascular improvements than others, which may be attributed to differences in initial fitness and metabolic profile, adherence to the prescribed intensity, or individual response to exercise. This variability suggests that while non-supervised FATmax training is effective for many, some individuals may benefit from additional support or adjustments to optimize their results.

### 4.4. Implications for Long-Term Health and Disease Prevention

The reductions in visceral fat and the improvements in cardiovascular function observed in this study underscore the potential of FATmax training as a preventive strategy against the progression of metabolic syndrome and CVD. Given that visceral fat is a significant contributor to systemic inflammation and insulin resistance, the ability of FATmax training to specifically target this type of fat makes it a valuable tool in managing the health risks associated with aging and obesity.

Moreover, the findings suggest that long-term adherence to FATmax training could contribute to sustained improvements in cardiovascular health, reducing the risk of future cardiac events. This study’s results are particularly relevant for middle-aged and older adults who may already exhibit early signs of metabolic dysfunction but are not yet candidates for more aggressive pharmacological treatments. By providing a non-invasive means of managing body weight and improving cardiovascular markers, FATmax training could help to delay the onset of more severe conditions, reducing the burden on healthcare systems.

## 5. Conclusions

A 6-month non-supervised walking program at FATmax is an effective strategy for reducing fat mass and improving cardiovascular function in obese individuals over 45 years old. The focus on a low to moderate exercise intensity that optimizes fat oxidation while maintaining a low cardiovascular strain makes FATmax training a promising intervention for reducing CVD risk in this demographic. The findings suggest that such a program is not only feasible for non-supervised application but also offers a sustainable pathway to improved health outcomes. Future research should examine the potential for personalized adjustments to maximize individual outcomes.

## Figures and Tables

**Figure 1 ijerph-22-00701-f001:**
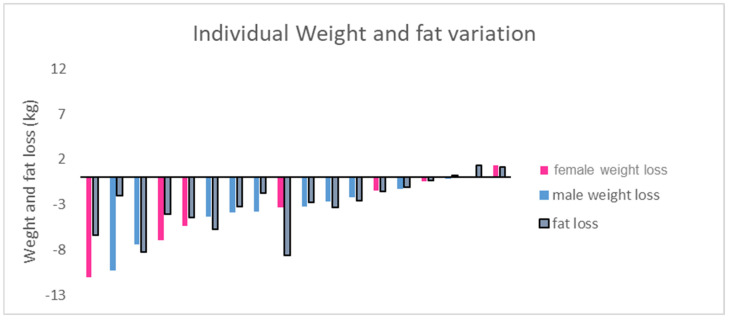
Individual weight and fat loss in participants, from beginning to end of study (6 months).

**Figure 2 ijerph-22-00701-f002:**
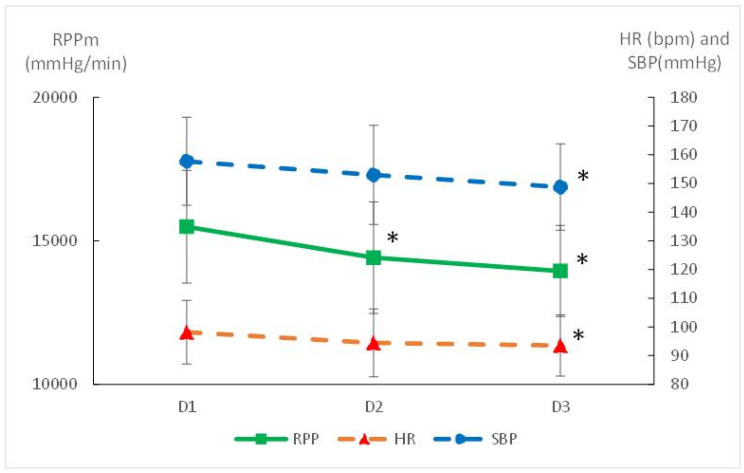
Mean cardiovascular response during exercise before (D1) and after 3 (D2) and 6 months of training (walking at FATmax). HR: mean heart rate; SBP: mean systolic blood pressure; RPP: mean rate of pressure product; * significantly different from D1 (*p* < 0.05).

**Figure 3 ijerph-22-00701-f003:**
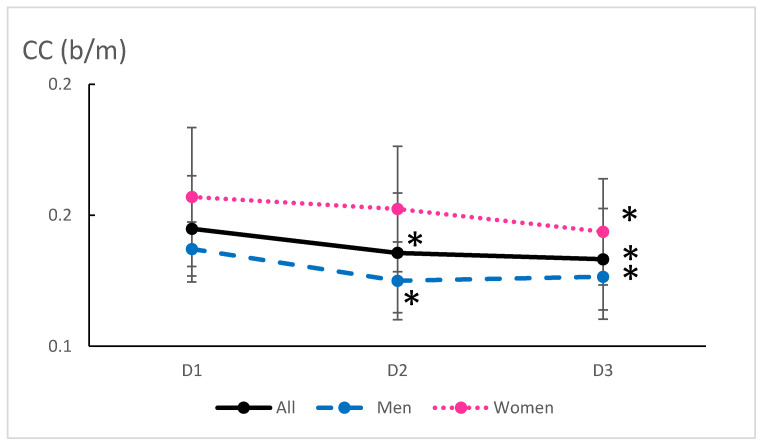
Mean cardiac cost during exercise before (D1) and after 3 (D2) and 6 months of training (walking at FATmax). CC: cardiac cost. * significantly different from D1 (*p* < 0.05).

**Table 1 ijerph-22-00701-t001:** Body parameters before (D1), and after 3 (D2) and 6 months (D3) of training (walking at FATmax). BMI: body mass index.

	D1	D2	D3	*p* (D1 vs. D3)	Effect Size
Body weight (kg)	95.3 (14.9)	91.4 (15.4)	91.6 (14.3)	0.032	0.25
female	85.9 (13.1)	83.1 (11.3)	82.0 (9.7)	0.089	0.34
male	101.3 (13.3)	96.7 (15.8)	97.7 (13.6)	0.004	0.27
BMI (kg/m^2^)	31.8 (3.8)	30.4 (4.1)	30.5 (3.5)	0.0002	0.36
female	31.5 (4.8)	30.3 (4.3)	29.9 (11.2)	0.019	0.40
male	32.0 (3.2)	30.5 (4.2)	30.8 (3.4)	0.03	0.30
Waist circumference (cm)	109.2 (9.0)	103.6 (9.9)	103.6 (7.4)	0.0003	0.64
female	108.7 (12.4)	104.4 (10.4)	101.9 (7.9)	0.013	0.65
male	109.4 (6.8)	103.0 (9.9)	104.8 (7.3)	0.050	0.64
% Fat	33.6 (7.1)	31.4 (8.3)	31.6 (7.2)	0.022	0.27
female	41.7 (3.3)	40.4 (2.8)	39.6 (1.2)	0.390	0.80
male	28.5 (2.0)	25.5 (4.6)	26.6 (3.7)	0.034	0.59
Lean body mass (kg)	63.6 (13.5)	62.7 (13.2)	62.8 (12.9)	0.512	0.06
female	49.9 (6.8)	49.3 (4.9)	49.5 (5.4)	0.259	0.07
male	72.3 (8.1)	71.3 (8.8)	71.4 (7.7)	0.096	0.01

**Table 2 ijerph-22-00701-t002:** Cardiovascular variables at rest and during exercise, before (D1) and after 3 (D2) and 6 months of training (walking at FATmax). Data are means ± SD. HR: heart rate; Qc: cardiac output; SBP: systolic blood pressure; DBP: diastolic blood pressure; RPP: rate-pressure product; SV: stroke volume.

	At Rest					During Exercise				
	D1	D2	D3	*p* (D1 vs. D3)	Cohen’s D	D1	D2	D3	*p* (D1 vs. D3)	Cohen’s D
HR (BPM)	77 (13)	72 (14)	73 (11)	0.097	0.36	98 (11)	94 (12)	94 (11)	0.022	0.42
Female	79 (15)	78 (17)	73 (13)	0.007	0.46	102 (14)	99 (13)	95 (12)	0.069	0.57
Male	76 (12)	68 (12)	73 (10)	0.059	0.27	96 (8)	91 (10)	93 (10)	0.327	0.30
SV (mL)	63.8 (12.1)	68.9 (18.3)	60.5 (17.4)	0.334	0.22	93.3 (17.6)	96.5 (19.3)	88.1 (18.9)	0.248	0.29
Female	60.0 (9.4)	62.5 (13.5)	55.0 (18.2)	0.432	0.35	82.0 (14.2)	84.3 (16.6)	79.8 (20.8)	0.368	0.13
Male	66.2 (13.4)	72.7 (20.3)	64.0 (16.8)	0.222	0.15	100.5 (16.1)	103.2 (17.9)	93.4 (16.5)	0.385	0.44
Qc (L·min^−1^)	4.9 (1.0)	4.8 (1.0)	4.3 (0.9)	0.245	0.60	9.1 (1.6)	9.0 (1.9)	8.3 (1.6)	0.353	0.49
Female	4.8 (1.2)	4.6 (0.8)	3.9 (1.0)	0.201	0.77	8.1 (1.7)	8.1 (1.1)	7.7 (1.9)	0.224	0.23
Male	5.0 (0.9)	4.9 (1.1)	4.6 (0.8)	0.205	0.47	9.7 (1.3)	9.4 (2.1)	8.7 (1.3)	0.436	0.78
SBP (mmHg)	131 (14)	130 (15)	123 (11)	0.012	0.59	158 (15)	153 (17)	149 (15)	0.001	0.57
Female	124 (11)	121 (12)	116 (11)	0.834	0.64	150 (13)	142 (10)	143 (16)	0.074	0.50
Male	135 (15)	135 (14)	127 (9)	0.168	0.65	162 (15)	160 (17)	152 (14)	0.407	0.66
DBP (mmHg)	85 (9)	86 (9)	81 (5)	0.014	0.54	96 (9)	95 (9)	90 (6)	<0.001	0.79
Female	80 (5)	81 (5)	78 (4)	0.175	0.48	91 (5)	89 (6)	86 (4)	0.092	0.99
Male	89 (10)	89 (9)	83 (5)	0.432	0.67	99 (10)	99 (9)	92 (6)	0.012	0.84
RPP (mmHg·min^−1^)	10,107 (1967)	9333 (1940)	8984 (1595)	0.004	0.28	15,493 (2412)	14,414 (2220)	13,949 (2348)	<0.001	0.62
Female	9844 (2360)	9479 (2470)	8486 (1843)	0.105	1.82	15,430 (3218)	14,075 (2242)	13,607 (2701)	0.02	0.61
Male	10,274 (1777)	9239 (1646)	9302 (1395)	0.183	0.59	15,533 (1917)	14,630 (2286)	14,166 (2204)	0.03	0.64
CC (beat·m^−1^)						1.45 (0.20)	1.36 (0.23)	1.33 (0.19)	<0.001	0.57
Female						1.57 (0.26)	1.52 (0.24)	1.44 (0.20)	0.020	0.56
Male						1.37 (0.10)	1.25 (0.15)	1.27 (0.16)	0.019	0.74

## Data Availability

The original contributions presented in this study are included in the article. Further inquiries can be directed to the corresponding author(s).

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
