# Peer review of "Cardiovascular, Hemodynamic, and Anthropometric Adaptations Induced by Walking Training at FATmax in Obese Males and Females over 45 Years Old"

_ijerph, 2025, doi:10.3390/ijerph22050701_

Round 1
Reviewer 1 Report
Comments and Suggestions for Authors
Dear Authors,
Thank you very much for your very interesting and insightful manuscript. Your study addresses an important aspect in the treatment of obesity—namely, the role of self-directed exercise training. Congratulations on this valuable contribution. I would like to offer a few comments and questions that may help to further improve the clarity and depth of the manuscript:
Abstract:
Line 20: Please review this sentence again—there may be a structural or grammatical issue, and it might not be fully complete.
- A general point: Would it be possible to explain why no control group without the training intervention was included? The inclusion of such a group could have strengthened the scientific validity of the findings. This must not be addressed in the abstract itself but rather the methods section.
Introduction:
Lines 79–81: It would be helpful to list concrete examples of the studies you refer to here to give the reader a clearer context.
- Additionally, are there studies on Nordic Walking that could be relevant in this context and might be worth mentioning?
Methods:
Could you provide further details on the exclusion criteria? Specifically, what does exclusion due to "regular activity" entail?
- Furthermore, could you elaborate on the rationale for not including a control group in the study design?
- How was adherence to the protocol monitored? For instance, did the participants keep an activity diary or were there other methods to ensure they did not engage in additional physical activity beyond the intervention?
Results/Discussion
- Line 167: It appears that a comma may be missing after "oxidation"—please check.
References:
- Kindly remove the remaining highlights or markings in the reference section to ensure a clean and professional presentation.
Author Response
Dear reviever,
Thank you very much for your comments which helped us to improve this manuscript. Please find our answers above ;

Reviewer 2 Report
Comments and Suggestions for Authors
It would also be of interest to include in lines 52-53 some data relating diastolic pressure to cardiovascular disease
"Could you please clarify the meaning of CCAS in line 94?"
In Study Design and Participants, please provide details of the sample selection process, how the sample size was estimated, and the participation rate for both men and women
From the skinfolds, include how you have estimated body fat (for example, did you use the Siri formula?) and include the reference
"In the statistical section, include which descriptive statistics you have used for both quantitative and qualitative variables. Since this is a study with few participants, effect size measures should be included."
"Non-significant p-values should also be included numerically, instead of using NS."
"Since you are using multiple comparisons, it would be advisable to apply the Bonferroni correction to the p-values."
Author Response
Thank you very much for your comments which helped us to improve this manuscript. Please find our answers above ;
